# The Diverse Role of CUB and Sushi Multiple Domains 1 (CSMD1) in Human Diseases

**DOI:** 10.3390/genes13122332

**Published:** 2022-12-10

**Authors:** Esra Ermis Akyuz, Sandra M. Bell

**Affiliations:** Division of Molecular Medicine, Leeds Institute of Medical Research, St James’s University Hospital, University of Leeds, Leeds LS9 7TF, UK

**Keywords:** CSMD1, cancer, complement system, schizophrenia, Parkinson’s disease

## Abstract

CUB and Sushi Multiple Domains 1 (*CSMD1*), a tumour suppressor gene, encodes a large membrane-bound protein including a single transmembrane domain. This transmembrane region has a potential tyrosine phosphorylation site, suggesting that CSMD1 is involved in controlling cellular functions. Although the specific mechanisms of action for CSMD1 have not yet been uncovered, it has been linked to a number of processes including development, complement control, neurodevelopment, and cancer progression. In this review, we summarise CSMD1 functions in the cellular processes involved in the complement system, metastasis, and Epithelial mesenchymal transition (EMT) and also in the diseases schizophrenia, Parkinson’s disease, and cancer. Clarifying the association between CSMD1 and the aforementioned diseases will contribute to the development of new diagnosis and treatment methods for these diseases. Recent studies in certain cancer types, e.g., gastric cancer, oesophageal cancer, and head and neck squamous cell carcinomas, have indicated the involvement of CSMD1 in response to immunotherapy.

## 1. Introduction

Research on the CUB and Sushi Multiple Domains 1 (*CSMD1*) gene, which is thought to be associated with tumour suppression and the immune system, has increased recently. Initially, the *CSMD1* gene was proven to be associated with schizophrenia (SZ), though CSMD1 is involved in many different mechanisms in the body with the discovery of its roles in the complement system, cancer, metastasis, cell migration, and epithelial–mesenchymal transition (EMT) [1,2,3]. Potentially, the *CSMD1* gene may help clarify the unknown pathologies of many diseases in the future. Consequently, a good understanding of the structure and possible metabolism/mechanism of the CSMD1 protein is important. In this review, we provide an overview of the general properties of the *CSMD1* gene and protein and its roles in various diseases.

## 2. *CSMD1* Structure

The CUB and Sushi Multiple Domains 1 (*CSMD1*) gene consists of 71 exons and spans a 2 MB DNA region on chromosome 8p23.2 [4]. *CSMD1* consists of 14 CUB domains separated by 14 complement control protein (CCP) domains at the N-terminus, followed by a tandem repetition of 15 CCP domains (Figure 1, [5]). The *CSMD1* gene encodes 17 splice variants in total [1]. The largest of these variations codes for a 3508-amino-acid protein with a 383 kDa molecular mass [4]. Conservative CUB domains, which span 110 amino acids, were first discovered in a variety of extracellular proteins, the majority of which were known to play a role in developmental processes and immunity, as well as in various cancer types [6,7]. Sushi domains are extracellular motifs that are frequently found in protein–protein interactions. Every Sushi domain has ∼60 amino acid residues with conserved tryptophan, glycine, proline, hydrophobic residues, and four invariant cysteines. Since many complement control proteins contain multiple Sushi domains, it is also referred to as a complement control protein [8,9,10]. The CSMD1 protein is a membrane-bound protein including a single transmembrane domain [4]. A putative tyrosine phosphorylation site inside this transmembrane region suggests that CSMD1 is involved in the regulation of cellular processes [11,12,13]. According to the latest data based on a combination of internally generated Human Protein Atlas RNA-Seq data and RNA-Seq data from the Genotype-Tissue Expression (GTEx) project, CSDM1 is predominantly expressed in the epithelial cells in many tissues with high expression in the brain, male reproduction system, and retina, followed by the breast and female reproductive system, spleen, and kidney [14,15].

## 3. *CSMD* Gene Family

CSMD1 is currently the best understood protein in the CSMD protein family, consisting of three structurally very similar proteins, with 14 CUB domains separated by a sushi domain, an additional uninterrupted array of sushi domains, a single transmembrane domain, and a short cytoplasmic tail. CSMD2 and CSMD3 are expressed at low levels in many tissues with the highest expression in the central nervous system. This expression pattern is similar to that of CSMD1, though CSMD1 expression is lower than that of the other two genes [16]. The striking similarity of all three *CSMD* genes brought up the question of whether CSMD2 and 3 are also tumour suppressors. A colorectal cancer study revealed that all three CSMDs were lower in colorectal carcinoma cells than in normal tissues [17].

*CSMD2* is located at the 1p34 chromosomal region, which may contain an oligodendroglioma suppressor, but its expression was found to be increased in some head and neck cancer cell lines [16]. Reduced CSMD2 expression was linked to tumour size, lymphatic invasion, and differentiation in colon cancer, the development of pulmonary sarcomatoid carcinoma, liver metastasis, and pancreatic cancer [17,18,19,20]. Furthermore, CSMD2 has a role in immature neuron development [21] and has also been identified as a risk factor for the development of schizophrenia (SZ) [22].

*CSMD3*, which is located on chromosome 8q23.3-q24.1, spans 1.2 Mb and encodes 73 exons [23]. CSMD3 expressed predominantly in the cortical neurons of the developing cortex was associated with neurodevelopmental disorders (NDDs) such as autism spectrum disorder (ASD) and SZ [24,25,26,27]. After elucidating the expression of CSMD3 in dendrites and its functions for dendrite formation [28], Song et al. proved that CSMD3 plays a critical role in early cortical neural network construction, which includes synaptogenesis and glia–vascular communication [29]. In addition, it was discovered that the *CSMD3* mutation is strongly associated with increased tumour mutation burden and poor clinical prognosis in ovarian cancer [30]. Furthermore, common *CSMD3* gene mutations were found in pulmonary carcinosarcomas [31] and oesophageal squamous cell carcinoma (OSCC) [32].

## 4. CSMD1 Function

CSMD1 has been implicated in a variety of processes including development, complement control, neurodevelopment disorders, and cancer progression, though specific mechanisms of action have yet to be identified (Figure 2, [5]) [3,33,34,35,36,37,38]. 

CSMD1 has been shown to modulate the SMAD pathway and has potential for intracellular signalling; due to its structural features, it may operate as a co-receptor or interact with growth factor receptors. It has also been reported that CSMD1 interacts with ECM components and goes through clathrin-mediated endocytosis (Figure 3, [5]) [39]. This review will focus on CSMD1 and its roles in the complement system, neurodevelopmental diseases, cancer and metastasis, and EMT.

### 4.1. CSMD1 in Complement System

The tandem Sushi domains at the C-terminus of CSMD1, a transmembrane protein composed of multiple CUB and Sushi repeats, have homology to the complementary activation of regulators (RCA) gene family [4,16,23,40]. Complement, a key part of the innate immune system, includes more than 30 proteins grouped in a proteolytic cascade, with effector mechanisms controlled by a variety of receptors [41]. Although complement is most plentiful in blood, all of its components can also be found in tissues due to diffusion or local transport [42]. Complement’s primary role is to recognise pathogens and damaged cells from an organism, induce inflammation, and destroy the pathogen’s cell membrane [43]. Complement is structured into three major pathways (classical, lectin, and alternative), each of which is inducted by sensory molecules that can detect pathogens or undesired material. This triggers a series of proteolytic activation processes that result in complement effector molecules such C3 fragments. The synthesis of the membrane attack complex (MAC), which can lyse Gram-negative bacteria and harm eukaryotic cells, is the last stage in the complement cascade [3]. The classical pathway is initiated by the binding of antibodies to antigens on the target cell. Complement C1s is activated as a result of this interaction, which is mediated via C1q activation. The C1s enzyme can cleave C4 and C2 components, resulting in the formation of C3 convertase (C4bC2b), which activates the complement cascade [41,44,45]. The alternative pathway stimulates the membrane attack complex by hydrolysing C3 molecules to produce a different C3 convertase (C3bBb). In this step, CSMD1 prevents C3 from being deposited on the cell surface, effectively shutting down the traditional complement pathway [37,40,46]. Consequently, CSMD1 was hypothesized to have a role in cell cycle regulation and apoptosis control due to its role on complement (Figure 4, [5]) [13].

### 4.2. CSMD1 in Neurodevelopment Diseases

The precise role of CSMD1 in immune responses awaits further elucidation, while a possible association of CSMD1 in autoimmune disease (neonatal lupus) has been reported [47]. In the light of current knowledge, we know that complement activity is tightly controlled in the brain and regulates C3/CR3-dependent axonal pruning by phagocytic microglia. During development, this mechanism maintains the perfect connection of neural circuits in the visual system [48]. Therefore, it is thought that complement may lead to abnormal synaptic elimination in other parts of the brain, which may affect the risk of both neurodegenerative and psychiatric disorders [49]. Furthermore, recent GWAS and SNP analysis have shown that *CSMD1* is closely related to neurological diseases such as SZ and Parkinson’s disease (PD) [50,51]. 

The discovery of a *Csmd1* promoter-associated lncRNA, which may be responsible for brain-specific promoter activity in the adult and developing central nervous system, added to the importance of Csmd1 for brain functioning [52]. Athanasiu et al. (2017) demonstrated that *CSMD1* variants are associated with immediate episodic memory and cognitive function [22]. Along with a GWAS study, it suggested that *CSMD1* is expressed significantly in the amygdala, which affects social behaviour, and CSMD1 can be used as a diagnostic marker for amygdala-related disorders [53].

In a study conducted with SZ patients, higher C4A mRNA levels were associated with more serious general psychopathology symptoms, while lower *CSMD1* mRNA levels were predicted to cause worse working memory [54]. Some *CSMD1* SNPs have been shown to be associated with cognitive ability in SZ patients [55]. It was also suggested that CSMD1 increased the risk of schizophrenia as a result of the negative effects of the A allele in *CSMD1* rs10503253 on brain activity [56]. A 2021 study by Abd El Gayed et al. found that the *CSMD1* mRNA expression and protein levels were significantly lower in SZ patients compared to the controls, suggesting that mRNA expression of the *CSMD1* gene may be a reliable and early diagnostic predictor of first-episode SZ in familial high-risk Egyptian children and young adults [57].

According to the results of whole-exome sequencing analysis performed on two Spanish families diagnosed with PD, it has been suggested that mutations in the *CSDM1* gene may cause the familial PD phenotype [58]. However, a study conducted on the Iranian population showed no association between *CSMD1* rs12681349 polymorphism and PD [59]. A case-control study conducted on the Han population in northern China in 2021 showed that polymorphisms in the *CSMD1* gene were closely related to PD, and significant differences were found in rs10503253 and rs1983474 polymorphisms between PD cases and controls. However, the researchers emphasized that studies in larger populations and other ethnic groups are needed to confirm the correlation between the *CSMD1* polymorphism and PD [51].

### 4.3. CSMD1 in Cancer

Abnormalities of the short arm of chromosome 8 have been associated with many carcinomas’ pathogenesis [60]. Toomes et al. and other scientists showed a hemizygous or homozygous deletion of the chromosome band at 8p23 encoding the CSMD1 protein in different cancer types such as oropharyngeal squamous cell carcinomas (OSCCs) and head and neck squamous cell carcinomas (HNSCCs) [1,61,62,63]. Although the function of CSMD1 is unclear, previous studies have shown that this transmembrane protein is engaged in a signalling cascade that regulates a variety of cell processes implicated in cancer formation such as proliferation and cell migration [4,13,64]. In this review, the differential analysis of CSMD1 expression between normal and cancer tissues was determined based on RNA sequencing data using tnmplot [65]. This analysis showed that CSMD1 expression was significantly increased in acute myeloid leukaemia (AML), liver, pancreas, skin, thyroid, and uterine corpus (EC) endometrial carcinoma compared to controls. However, adrenal, bladder, breast, colon, ovary, prostate, rectum, renal chromophobe (CH) cell carcinoma, renal clear cell carcinoma (CC), renal papillary cell carcinoma (PA), and lung adenocarcinoma (AC) had less CSMD1 expression compared to controls (Figure 5, [65]). 

Previously, we identified reduced CSMD1 expression in 79/275 (28.7%) of invasive ductal breast cancer patients, which were associated with high tumour grade and poor overall survival. More importantly, CSMD1 was also an independent predictor of overall survival [66]. An mRNA study by Escudero-Esparza et al. in 127 breast cancer samples also found that low levels of CSMD1 expression was associated with statistically significant lower survival compared to those with high levels [67]. In a later study, we created a three-dimensional culture model of MCF10A cells with decreased CSMD1 expression that indicated that the reduction of CSMD1 expression resulted in the formation of bigger and less differentiated breast ductal structures [35]. A recent deep whole-genome sequencing study identified CSMD1 deletions in 50% of breast cancer patient-derived xenografts, suggesting a role in driving aggressive breast cancer. The low expression of CSMD1 was also associated with reduced survival in the breast cancer METABRIC study [68]. Gialeli et al. suggested that CSMD1 provides tumour suppression by interacting with the EGFR pathway and can be used as a biomarker for predicting chemotherapy response in highly invasive breast cancer [69]. Additionally, Tang et al. discovered that the decreased expression of CSMD1 in melanoma cells has a lower influence on melanoma cell migration and proliferation, and that CSMD1 can serve as a tumour suppressor gene in melanoma cells [13]. Furthermore, Zhang et al. discovered that CSMD1 expression in colorectal cancer is low, is linked to overall survival, can be utilised as a predictor of colorectal cancer, and plays a vital part in the prognosis of the disease [17]. Additionally, reduced *CSMD1* gene expression was associated with poor prognosis in HNSCC and prostate cancer [38,70]. A further HNSCC study interrogating three publicly available genome-wide expression datasets found CSMD1-inactived cancers demonstrated a reduced prognosis [12]. 

Similarly, Fan et al. showed that the *CSMD1* mutation status was an independent predictor of prognosis in oesophageal cancer patients. This study also found CSMD1 wild-type oesophageal cancer patients were more susceptible to paclitaxel chemotherapy [71]. It has shown that *CSMD1* is dramatically downregulated in gastric cancer (GC) tissues compared to normal tissues via RT-PCR, and the overexpression of microRNA-10b, a direct target of CSMD1 in GC cells, was found to increase GC cell vitality, migration, and invasion [72,73,74]. Another study discovered that miR-642b-3p acts as an oncomiR that promotes tumour progression in GC by repressing CSMD1 expression and inactivating the Smad signalling pathway, which could contribute to the development of potential therapies for GC treatment [75].

A study on hepatocellular carcinoma (HCC) identified lncCSMD1-1 upregulated in HCC and directly binding to the MYC protein in the nucleus of HCC cells, promoting HCC progression and the upregulation of MYC protein. Furthermore, lncCSMD1 has been shown to act as an oncogene, promoting HCC cell proliferation, migration, invasion, and EMT. As a result, it was speculated that lncCSMD1 could be a novel and reproducible prognostic biomarker for HCC patients, as well as playing an important role in HCC progression [76]. Using the GEO and TCGA databases, a study identified several novel driver genes including *CSMD1* associated with HCC and demonstrated that this gene was strongly related to the prognosis of early recurrence and an effective prognostic marker for HCC [77].

### 4.4. CSMD1 in Metastasis and EMT

Metastasis is a complex process that includes events such as loss of adhesion, increased invasiveness, and motility to perform intravasation, joining the circulation through lymph nodes and blood vessels, and connecting to blood vessels [78]. Tumour cells transform from an epithelial phenotype to a mesenchymal phenotype to acquire invasive characteristics, and metastasis is one of the key points in the formation of this epithelial–mesenchymal transition (EMT). Significant evidence indicates that the tumour suppressor gene *CSMD1* plays an important role in metastasis. Deletions of the CSDM1 locus chromosome 8p23 appear to be clinically important because an allele deficit at this coding region has been linked to poor prognosis, recurrence, and metastasis in many cancer types [79]. Previously, we demonstrated that CSMD1 knockdown using short hairpin RNA in three different cell lines (breast MCF10A, prostate LNCaP, and metastatic MDA-MB-435) resulted in striking morphological changes similar to those previously reported in EMT. Furthermore, functional assays on the MCF10A breast cancer line found loss of CSMD1 expression enhanced migration and invasion while reducing adhesion to Matrigel and fibronectin [35].

In melanoma cells, an increased expression of CSMD1 reduces the rate of cell migration [13]. In a xenograft model of human breast cancer, mice injected with MDA-MB-231 breast cancer cells, either alone or with artificially increased CSMD1 expression, found that there was a significant reduction in lung metastasis in the CSMD1-expressing group when compared to the controls. This study also demonstrated that the overexpression of CSMD1 in breast cancer cell lines BT20 and MCF7 inhibited migration, adhesion, and invasion [67].

EMT is a normal biological process driving mammary gland development. In order to investigate the association between CSMD1 and EMT further, mammary gland development during puberty was studied in a Csmd1 knockout (KO) mouse model. Our study identified increased ductal development during the early stages of puberty in the KO mice, characterised by increased ductal area and terminal end bud number at 6 weeks. The increased expression of various proteins (Stat1, Fak, Akt, Slug/Snail, and Progesterone receptor) was identified in the Csmd1 KO mice mammary glands at 4 weeks, followed by lower expression levels from 6 weeks in the KO mice compared to the wild-type control mice. This study found an association between Csmd1 and cell invasiveness regulation, which might be regulated by changes in cell adhesion processes (Figure 6, [5]) [33]. Our study indicates a novel role for Csmd1 in mammary gland development, with Csmd1 KO causing significantly more rapid mammary gland development, suggesting an earlier adult mammary gland formation [33].

### 4.5. CSMD1 Inactivation

There are many different ways that CSMD1 might be dysregulated, including epigenetic silencing, *CSMD1* gene deletions and mutations, and microRNA interference. Interrogation of the Cancer Genome Atlas (TCGA) has identified alterations in CSMD1 in many malignant tumours such as breast cancer (~5%), prostate cancer (5.7%), bladder urothelial carcinoma (8%), lung cancer (7%), ovarian cancer (7%), liver cancer (7%), and colorectal adenocarcinoma (7.4%) (Figure 2). It was discovered that miR-10b reduces the expression of CSMD1 and that it is substantially expressed in HCC compared to healthy tissue [64]. Similarly, the downregulation of CSMD1 in glioblastoma was associated with the overexpression of miRNA-10a and miRNA-10b [80]. MiR-137, which was discovered through genome-wide association studies in SZ, is another microRNA that specifically targets CSMD1 [81]. Both miR-10b and miR-137 have been found to bind to the 3′ untranslated region of the *CSMD1* gene [80,81].

### 4.6. CSMD1 and Immunotherapy

Recently, several studies have suggested that CSMD1-inactivated cancers may respond to immune checkpoint inhibitors [12,71,72,73]. An oesophageal cancer study found CSMD1 mutated cancers were associated with a high tumour mutation burden (TMB). As expected, the high TMB correlated with high expression of PD-1, suggesting that these patients may benefit from PD-1 inhibitor immunotherapy [71]. Furthermore, a large-scale genomic study on HNSCC cancers demonstrated an association between CSMD1-inactivation and tumour immunity [12]. Similarly, a GC study revealed that patients with *CSMD1* mutations had significantly higher TMB and better prognoses than *CSMD1*-wild patients. Following that, it showed that several immune-related signalling pathways were upregulated in the *CSMD1*-mut samples; that there was a higher proportion of anti-tumour immune cells, such as CD4+ Th1 cells, NK cells, M1 macrophage cells, and PDC; a lower proportion of tumour-promoting immune cells, such as Treg cells, M2 macrophage cells, and endothelial cells; and that PD-L1 was upregulated [73]. Another GC study confirmed the association between CSMD1 mutation with TMB and high PDL1 expression and increased survival supporting the potential of CSMD1 as a biomarker for assessing immune checkpoint inhibitor therapy in GC patients [74].

## 5. Conclusions

Elucidating the relationship of CSMD1 with the aforementioned diseases in this review may be important in terms of both treatment and diagnosis in the future. Recent studies have revealed a possible role for CSMD1 in immunotherapy for some cancer types [73,74,75,76,77,78,79,80,81,82]. The determination of the factors affecting the expression of CSMD1 and the clarification of the signalling pathways it affects will increase the clinical utility of CSMD1.

## Figures and Tables

**Figure 1 genes-13-02332-f001:**
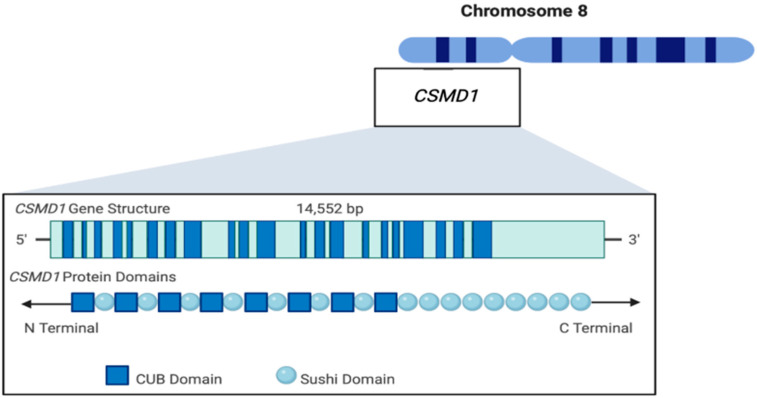
Schematic illustration of the CUB and Sushi Multiple Domains 1 (*CSMD1*) gene and protein structures.

**Figure 2 genes-13-02332-f002:**
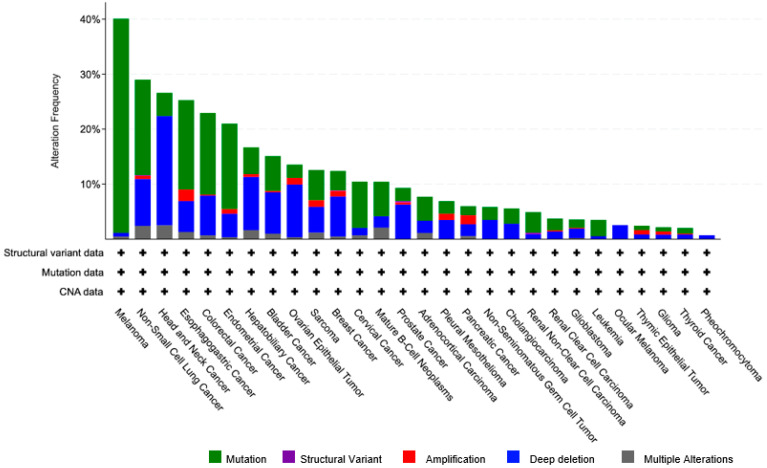
Frequency and type of genetic alterations at the *CSMD1* locus in different cancer types.

**Figure 3 genes-13-02332-f003:**
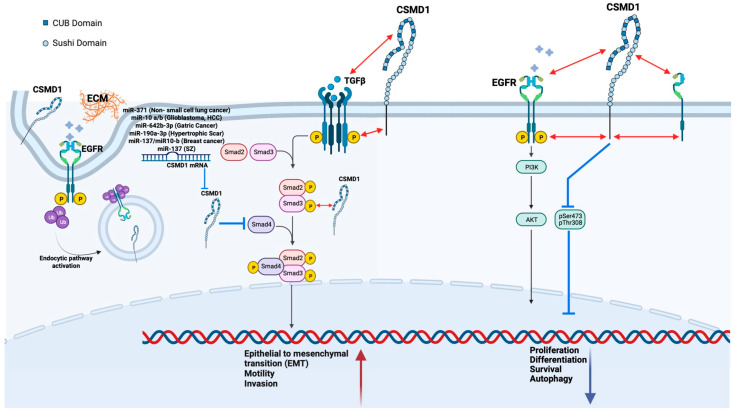
Summary of CSMD1-mediated functions.

**Figure 4 genes-13-02332-f004:**
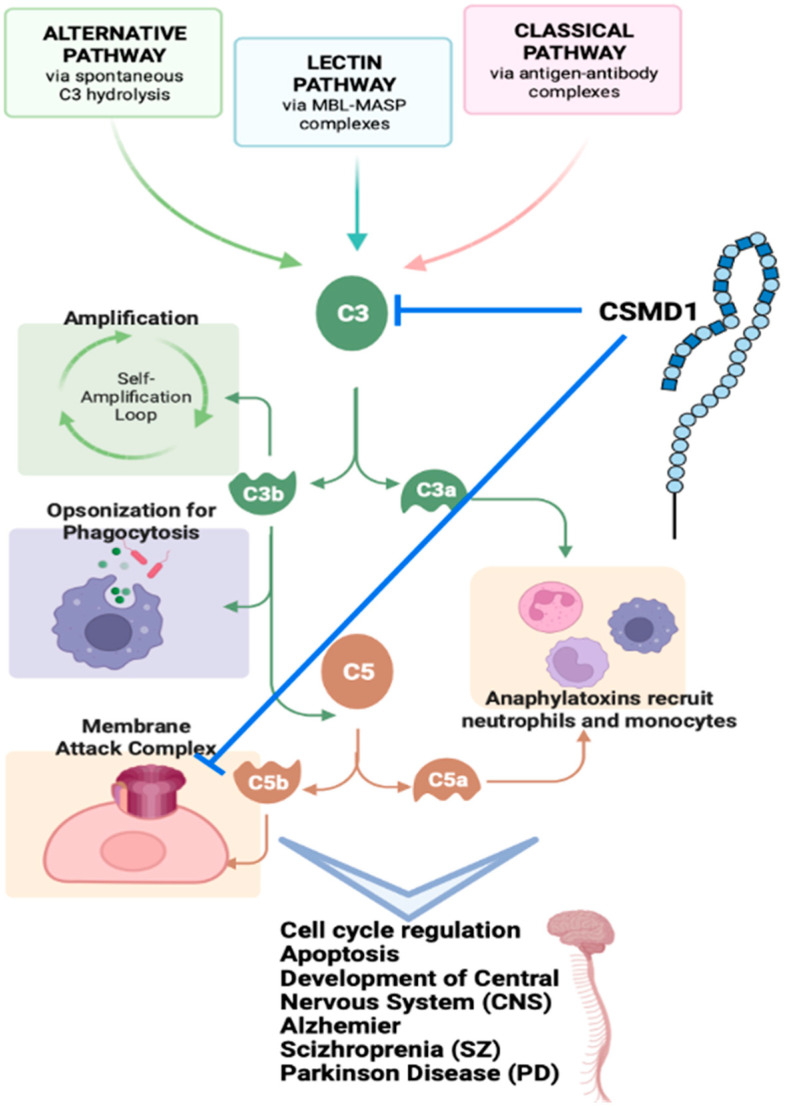
Overview of complement system and CSMD1.

**Figure 5 genes-13-02332-f005:**
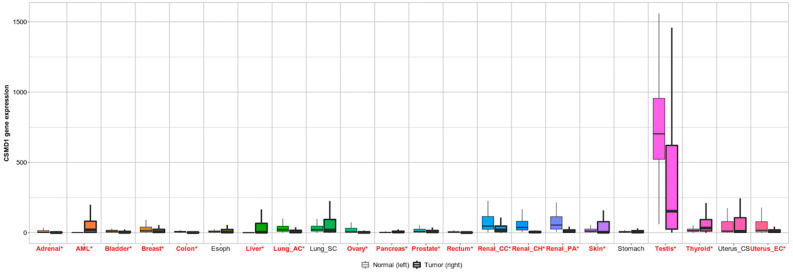
*CSMD1* differential gene expression in normal and tumour tissues. *CSMD1* is overexpressed in a number of tumour types compared to normal tissue, with AML and liver being the highest. However, *CSMD1* expression is not strongly observed in renal and prostate tumours. A Mann–Whitney U test was performed to mark the significant difference in expression between normal and tumour samples depicted by red colour.

**Figure 6 genes-13-02332-f006:**
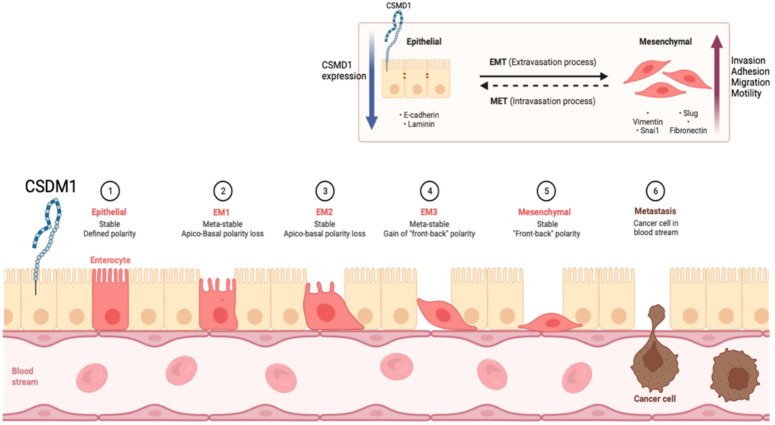
Brief summary of EMT, MET, and CSMD1.

## Data Availability

All relevant data have been cited within this review article.

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
