# Peer review of "The Diverse Role of CUB and Sushi Multiple Domains 1 (CSMD1) in Human Diseases"

_genes, 2022, doi:10.3390/genes13122332_

Round 1
Reviewer 1 Report
Review is well-written and informative. Figures are well-designed. Although authors mainly used the recent references, it will be better to change some older ones.
This is a very interesting contribution to public health; On the other hand, this work presents some corrections according to the following comments:
1. It could be better to text “in human diseases” instead of “in human disease” in the main title.
2. In abstract section, line 13, “we summarise CSMD1 functions in the complement system, schizophrenia, Parkinson’s disease, cancer, metastasis, and Epithelial mesen-chymal transition (EMT)”; diseases and cellular processes should be written separately.
3. There are some repeated sentences such as “CSMD1 is a putative tumour suppressor gene that maps to human chromosome 8p23”. It has already mentioned before. It could be better to delete these kinds of repeats.
4. In “CSMD1 in Neurodevelopment Diseases” section, it may be better to make a different paragraphs for each of the diseases.
5. In “CSMD1 in Cancer” section in line 167, authors texted that expression of CSMD1 significantly decreased in GBM compared to controls. After that, in line 170, they mentioned CSMD1 had less expression in GBM. I did not get well which one is right. Another point is I could not see the expression of CSMD1 in GBM in Figure 5.
6. Authors are really careful while they texted the gene names according to nomenculature but there are some typing errors in lines 205, 213 and 215. CSMD1 should be written italic in these lines.
Reviewer 2 Report
In the review manuscript titled "The diverse role of CUB and Sushi Multiple Domains 1 (CSMD1) in human disease" Akyuz and Bell describe the last discovery about the role of CSMD1 in human disease in general, with a particular focus on cancer progression. Although it could represent a good starting point for all the researchers start focusing on this protein, to have an overall introduction in the field, the manuscript would have benefited by the presence of a paragraph dedicated to the role of CSMD1 in immunotherapy, which represent a hot topic in clinics.
